# Cross-Cultural Adaptation and Validation of the Arabic Version of the Prolapse Quality of Life Questionnaire in the United Arab Emirates

**DOI:** 10.3390/healthcare12040444

**Published:** 2024-02-08

**Authors:** Asma Abdelrahman Alzarooni, Tamer Mohamed Shousha, Meeyoung Kim

**Affiliations:** 1Department of Physiotherapy, College of Health Sciences, University of Sharjah, Sharjah P.O. Box 27272, United Arab Emirates; u19102443@sharjah.ac.ae (A.A.A.); tshousha@sharjah.ac.ae (T.M.S.); 2Department of Physiotherapy, Kalba Hospital, Sharjah P.O. Box 11195, United Arab Emirates; 3Neuromusculoskeletal Rehabilitation Research Group, Research Institute of Medical and Health Sciences, University of Sharjah, Sharjah P.O. Box 27272, United Arab Emirates; 4Department of Physical Therapy for Musculoskeletal Disordered and Its Surgery, Faculty of Physical Therapy, Cairo University, Giza 12511, Egypt; 5Healthy Aging, Longevity and Sustainability Research Group, Research Institute of Medical and Health Sciences, University of Sharjah, Sharjah P.O. Box 27272, United Arab Emirates; 6University of Sharjah Center of Excellence for Healthy Aging, Sharjah P.O. Box 27272, United Arab Emirates; 7Laboratory of Health Science & Nanophysiotherapy, Department of Physical Therapy, Graduate School, Yongin University, Yongin 17092, Republic of Korea

**Keywords:** quality of life, psychometrics, cross-sectional, prolapse

## Abstract

Background: Given the extensive translation of the Prolapse Quality of Life Questionnaire (P-QoL) into many languages, it is imperative to develop an Arabic version to facilitate the study of pelvic organ health within the Arabian culture. Objective: The aim of this study was to investigate, cross-culturally adapt, and validate the Arabic version of the P-QoL. Study Design: This study involved cross-cultural adaptation and psychometric testing. Methods: A total of 90 participants were included in the study. This cross-sectional study was carried out in two phases; during phase I, the P-QoL was translated and adapted from English into Arabic. The Arabic version was psychometrically validated during phase II using the test–retest reliability and internal consistency with Cronbach’s alpha coefficient, convergent construct (CC) validity among the four study tools using Spearman’s coefficient (*r*), and discriminative validity using Mann–Whitney test to find the differences between the means of the two samples. Results: A satisfactory level of semantic, conceptual, idiomatic, and content comparability was reached in the cross-cultural adaptation of the Arabic version of the P-QoL. The internal consistency was high in terms of psychometric validation, with a Cronbach’s alpha coefficient of 0.971 for the P-QoL. The test–retest results showed high reliability, with the interclass correlation coefficient (ICC) of the P-QoL determined as 0.987. The convergent construct validity was highly acceptable (moderately strong), reflecting a positive correlation between the Arabic version of the P-QoL and the Australian Pelvic Floor Dysfunction Questionnaire (APFD) (*r* = 0.68; *p* < 0.001). Similarly, a significant convergent validity of the Arabic version of the P-QoL and the Visual Analogue Scale (VAS) (*r* = 0.47; *p* < 0.001) was observed, as well as a correlation between the APFD and the VAS (*r* = 0.46; *p* < 0.001). However, there was no significant correlation between the 12-Item Short-Form Survey (SF-12), the P-QoL, the APFD, and the VAS. Conclusion: Based on the significant correlation found between the Arabic APFD and the VAS, the results reveal good reliability, internal consistency, and construct validity. It is recommended that Arabic-speaking females with pelvic organ prolapse use the Arabic version of the P-QoL. More research is needed to assess the responsiveness of the P-QoL.

## 1. Introduction

Urinary incontinence (UI), fecal incontinence (FI), and pelvic organ prolapse (POP) are all common and disabling conditions that have a significant impact on one’s quality of life (QoL) [1]. There are many urogenital signs that affect 23–49% of females, and it is predicted that there will be 43.8 million cases with urogenital signs by 2050 in undeveloped and developing countries. This will bring negative emotional and physical effects on their quality of life [2].

According to previous research, almost 25% of all women in the United States of America suffer from pelvic floor disorders (PFDs), and 20% of these women will need UI or POP surgery at some point in their lives [3].

As a definition, the backward descent of female pelvic organs, namely the uterus, bladder, and/or rectum, through or into the vagina is referred to as POP. POP affects 20 to 50 percent of women worldwide, and the risk increases with age, parity, and heavy lifting [4].

In addition to the underlying anatomical conditions, it is essential to consider a female’s general pelvic performance [3]. A standardized questionnaire plays a key role in the identification of a condition’s signs and helps clinicians to accurately determine and characterize any symptom [5]. To accomplish this, in 2003, condition-specific QoL instruments were developed and published in Italy by Digesu et al. [6]. It was then translated into several languages, including English, German, Dutch, Slovakian, Persian, Portuguese, Thai, Japanese, Amharic, and Turkish [4,6,7,8,9,10,11,12,13,14].

The Prolapse Quality of Life Questionnaire (P-QoL) is one of the few validated and reliable condition-specific questionnaires designed to measure the impact of urogenital prolapse on patients’ QoL. The questionnaire addresses overall health, prolapse effect, role restrictions, physical limits, social limitations, relationships, emotional difficulties, sleep/energy abnormalities, and prolapse severity [14]. 

Using instrumental tools in a patient’s native language in the assessment process allows for the clear identification of the condition, reflecting all life aspects [15]. A large majority of pelvic floor problem surveys were first written in English. However, because many of the symptoms may be experienced and reported differently by women from different cultures, an effort should be made to establish culturally relevant questionnaires. As a result, the cross-cultural adaptation and validation of relevant surveys in Arabic among non-English-speaking communities is necessary [16].

Elbiss et al. conducted a study reporting the prevalence of pelvic organ prolapse in the United Arab Emirates (UAE), in which they confirmed that the UAE, as in all other Arab countries, has a high level of parity. They also concluded that POP symptoms were common in Emirati women. Independent risk variables were a history of chronic constipation, chest illness, education level, work type, birth weight, and BMI. More healthcare efforts were needed to educate the public about these risk factors [17]. Therefore, establishing an Arabic-translated version of the P-QoL would be beneficial.

The cultural adaptation and validation of the questionnaire involves cross-cultural adaptation, followed by psychometric property validation. This approach, in addition to being less expensive than developing new questionnaires, helps researchers to conduct cross-national comparisons. The aim of the cross-cultural adaptation of questionnaires is to ensure that there is semantic, logical, idiomatic, and material equivalence with the original source, which necessitates a systemic approach [18]. Meanwhile, the aim of validation is to prove the reliability and validity of the Arabic version of the P-QoL as a measurement scale.

To date, no valid Arabic version of the P-QoL has been developed in the UAE or any other Middle Eastern country, despite its necessity. Indeed, in the future, the questionnaire may be implemented in outpatient facilities to aid in identifying the problems that patients have.

## 2. Materials and Methods

This study was carried out in two phases. During phase I, the P-QoL was translated and culturally adapted from English into Arabic. The Arabic version was psychometrically validated during phase II. This study was approved by the University of Sharjah Research Ethics Committee (REC-21-06-23-02-S) and the Ministry of Health and Prevention Research Ethics Committee (MOHAP/DXB-REC/SSN/No. 94/2021), UAE. The entire procedure of translation and validation is shown in Figure 1.

### 2.1. Participants

This study was delimited to Emirati females who spoke Arabic as their native language, aged between 18 and 65, and who were diagnosed with POP with or without symptoms and were willing to participate in the study voluntarily. Women were classified as symptomatic if they had signs of uterovaginal prolapse (such as a genital bulge or a feeling of heaviness in the vagina) and as asymptomatic if they did not have any symptoms. Women in the asymptomatic population had routine check-ups, including pelvic ultrasound scans, to check their complaints regarding menstrual pain, heavy cycles, endometriosis, amenorrhea, the need for abortion, breast cancer, contraception, or other non-prolapse disorders [19]. Study group classification into symptomatic and asymptomatic was performed after gynecological screen testing.

Exclusion criteria included pregnant women, patients who recently underwent surgery or childbirth, women with acute symptoms of urinary tract infection, women unable to read, and non-Arabic speakers.

During the period between March 2022 and July 2022, recruitment was carried out across gynecology outpatient clinics in Emirates Health Services (EHS), following the ethical approval received from the Research Ethics Committee of the University of Sharjah and the Research Ethics Committee of the Ministry of Health and Prevention, UAE.

### 2.2. Prolapse Quality of Life Questionnaire (Study Tool)

Nine sections, with a total of sixteen questions, constitute the unique, multidimensional P-QoL. The P-QoL was initially developed to assess how women’s QoL was affected by POP. The first question focuses on women’s overall health. The second question evaluates the impact of POP on women’s quality of life. The third and fourth questions assess the limitations on daily activities that the urogenital prolapse may impose. The next four questions evaluate any potential physical and social constraints resulting from POP. The ninth, tenth, and eleventh questions examine how women’s personal relationships are affected by their uterovaginal prolapse. The effects of the urogenital prolapse on women’s emotions are assessed in questions 12 through 14.

Questions 15 and 16 evaluate symptom severity and the impacts of POP on sleep and energy [20].

### 2.3. Phase I: Translation and Cultural Adaptation

After obtaining permission from the developers, we followed a standard procedure according to the internationally accepted guidelines for translation and adaptation [21]. The original English form was forward-translated by two native-speaking Arabic translators, who were also fluent in English. They translated the document from English into Arabic. The two translators (one had a medical background, and the other was an official translator with a non-medical background) worked separately to translate the English version of the P-QoL into Arabic. In the second step, the translators synthesized the two Arabic translations into a single translation. A third translator performed a backward translation (into English) of the Arabic version generated in the second step. The original text and the back-translated version were then carefully evaluated to identify and address any potential contradictions. The third translator concurred that the original material had been preserved without any ambiguity. No modifications were included in the translated version, and the Arabic translation initially generated in the second step remained unchanged.

### 2.4. Phase II: Psychometric Validation

Prior to the main study, a pilot study was conducted on 30 randomly chosen participants with the help of simple random sampling using the lottery method. Participants who completed the Arabic P-QoL 2 times with a time gap of at least 48 h were included to evaluate test–retest reliability.

Out of 156 participants, only 30 participants were chosen for the pilot study, while 31 respondents were excluded from the study due to incomplete responses, and 90 participants successfully completed the study. The 90 participants completed Arabic P-QoL, the Australian Pelvic Floor Questionnaire (APFD), the Visual Analogue Scale (VAS), and the SF-12 to measure convergent construct validity. To measure internal consistency, the Arabic version of the P-QoL was compared to the APFD, which is a validated, dependable tool that is used in clinics to evaluate all aspects of PFD signs and severity and provides clinicians with quick and accurate QoL information [3]. While there are several questionnaires available to determine the signs of PFD, their severity, and their effect on QoL, they do not cover all elements of the disorder (bowel, bladder, prolapse, and sexual dysfunction). However, the APFD is a reliable and valid means of assessment [3].

### 2.5. Sample Size

Using the G-Power software (version 3.1.9.7) [22], to have an expected Cronbach’s alpha of 0.8, a significance level (α, two-sided) of 0.05, and a power of 0.9, 78 participants were required for this study. Considering an excess of 10% for possible dropouts, the minimum requirement was 86 participants. After the initial screening, 121 participants remained in this study. A total of 90 participants successfully completed our study.

### 2.6. Statistical Analysis

Descriptive statistics were performed to determine the means and standard deviations for sociodemographic features and specific medical history data, which were used as continuous variables, namely age, the pelvic organ prolapse quantification system (POP—Q stages), and type of delivery, while the factors associated with frequency and proportion were used to describe categorical variables, namely parity, education status level, and body mass index (BMI). Before analytical processing, uncompleted responses were filtered and removed. Floor and ceiling effects were considered if more than 15% of participants achieved the lowest or highest scores, respectively [23]. A *p*-value of <0.05 was considered statistically significant.

All statistical analyses were performed using the Statistical Package for Social Sciences (SPSS) program (version 26.0, 2019, New York, NY, USA). Baseline demographic data were analyzed using Fischer’s exact test. The test–retest reliability was evaluated with the intraclass correlation coefficient (ICC). Reliability was examined at 95% confidence interval (CI) levels. The absolute measurement error calculated by the smallest detectable change (SDC), also known as the minimum detectable change (MDC 95%) [24], was obtained by determining the standard error of measurement (SEM).

Accordingly, internal consistency was measured by means of Cronbach’s alpha. The evaluation of the convergent construct validity and correlations was carried out using Spearman’s coefficient (*r*).

## 3. Results

A satisfactory level of semantic, conceptual, idiomatic, and content comparability was reached in the cross-cultural adaptation of the Arabic version of the P-QoL. A total of 126 participants were recruited across EHS hospitals with and without symptoms of POP. Among the 126 Emirati women who were invited, 121 agreed to participate in the study; 31 respondents were excluded from the study due to incomplete responses, and 90 participants successfully completed the study (Figure 2).

Out of the 90 women involved in the main study, 61.11% (55) expressed symptoms of pelvic organ prolapse, whereas 38.89% (35) presented no affecting symptoms.

Table 1 demonstrates the baseline sociodemographic and medical characteristics of the study’s participants, with a mean age of 42.53 for the symptomatic group and 33.54 for the asymptomatic group. Table 2 presents the results for test–retest repeatability and internal consistency. The study findings demonstrated a strong level of internal consistency for the P-QoL measure, as shown by a Cronbach’s alpha coefficient of 0.971. The interclass correlation coefficient (ICC) for the P-QoL measure was 0.987, indicating a high level of reliability.

Additionally, Cronbach’s alpha coefficients for the subscales of role limitations, physical limitations, social limitations, personal relationships, emotions, and severity measures ranged from 0.936 to 0.836, demonstrating high reliability. However, the sleep and energy domains were found to have a lower, yet still acceptable, level of reliability, with a coefficient of 0.682.

Table 3 reports the absolute reliability of the measures, standard error of measurement, and minimal detectable change [25]. Table 4 presents the convergent construct validity (Spearman’s coefficient (*r*)) between the P-QoL, the APFD, the SF-12, and the VAS. The convergent construct validity was highly acceptable (moderately strong), reflecting a positive correlation between the Arabic version of the P-QoL and the APFD (*r* = 0.68; *p* < 0.001), and a significant correlation was observed between the convergent construct validity (CC) of the Arabic version of the P-QoL and the VAS (*r* = 0.47; *p* < 0.001). There was a correlation between the APFD and the VAS (*r* = 0.46; *p* < 0.001), whereas there was no significant correlation between the SF-12 and the P-QoL, the APFD, and the VAS.

A further detailed statistical correlation was determined for each subtitle domain of the study tools. Table 5 presents the convergent construct validity (Spearman’s coefficient (*r*)) for each instrument’s subscale between the P-QoL, the APFD, the SF-12, and the VAS. In order to perform a comprehensive evaluation, the correlation between the convergent construct validity of the Arabic P-QoL subscales and each instrument was determined. A positive moderate-to-high correlation was observed between general health perceptions (GH), prolapse impact (PI), role limitations (RL), physical limitations (PL), social limitations (SL), personal relationships (PR), emotions, sleep/energy, severity measures (SM), and the APFD, ranging from *r* = 0.443 (*p* < 0.001) to *r* = 0.667 (*p* < 0.001). By contrast, in the APFD subscales, the APFD bladder function exhibited the highest correlation with the P-QoL subscales, ranging from *r* = 0.595 (*p* < 0.001) to *r* = 0.854 (*p* < 0.001). Similar to the P-QoL items, there was a positive moderate-to-high correlation with the VAS. Conversely, there was no correlation between the P-QoL subscales and the SF-12.

The Mann–Whitney test was used to find the discriminative validity between the symptomatic and asymptomatic groups (Table 6). The significance value of the Mann–Whitney test for the effect of pelvic organ prolapse was *p* < 0.001, revealing a statistically significant difference between the mean scores of the two groups in the P-QoL, the APDF, and the VAS. The worst POP effects were observed in the symptomatic group. In the confirmatory factor analysis, the CFI was 0.74, indicating an acceptable fit. The Kaiser–Meyer–Olkin measure of sampling adequacy (KMO) was 0.869, indicating that the sampling was adequate with a *p*-value of 0.000. X2/df was 2.29, and the root mean square error of approximation (RMSEA) was mediocre, with a value of 0.12.

## 4. Discussion

The aim of our study was to validate and culturally adapt the Arabic version of the P-QoL as it is a valid and reliable tool for assessing POP. There are currently a limited number of tools in the Arabic language to assess the effects of pelvic organ prolapse on quality of life. Therefore, the validation of the Arabic version of the P-QoL has a great importance considering P-QoL provides analysis of the symptoms affecting the quality of life of women with POP.

The APFD is a valid and reliable tool that has been translated into multiple languages, including Arabic [3,26], which was used as a comparison tool along with the SF-12 and the VAS. The selection of the APFD was based on the congruence between the dimensions it measures with the P-QoL. In this study, the Arabic version of the P-QoL was validated to establish a medical health-related quality assessment of Arabic-speaking females expressing symptoms of POP. This validated arabic P-QoL can be used as a screening test and might help developing a treatment plan in the clinics.

The translation/back-translation process was utilized for the cultural adaptation of the Arabic version of the P-QoL in an identical way to how the Amharic version of the P-QoL was translated [4], involving a dual translation process with three steps of translation, cultural adaptation, and psychometric validation. The pilot investigation demonstrated that the P-QoL was effective, albeit with some small adjustments required when finalizing the local language version to enhance the technical similarity between the English and Arabic versions.

In our study, the Arabic version of the P-QoL demonstrated great reliability and construct validity, indicating excellent agreement with previous studies [4,6,7,8,9,10,11,12,13,14]. The reliability was tested using test–retest reliability and internal consistency with the ICC and Cronbach’s alpha. The internal consistency was high, and the Cronbach alpha scores had a high agreement. The ICC values showed high reliability, indicating a very good-to-excellent agreement. For further accuracy and to report absolute reliability, the SEM and MDC were calculated. Cronbach’s alpha coefficient of test–retest reliability was greater than 0.8, similar to the Persian [18], Spanish [27], and Polish [28] versions. The sleep/energy domain had a Cronbach’s alpha coefficient of 0.7 in the Persian version. In our Arabic version, the severity measures domain had a Cronbach’s alpha coefficient of 0.7, which was similar to Slovakian [10] and Portuguese [11] studies. These variations among studies reflect the cultural differences and include difference in sampling methods.For example, in the Persian version, the inclusion criterion was volunteers aged 60 or older. Despite the fact that none of the previous studies on the validation of the P-QoL used the MDC as an additional reliability measure, we preferred using it to record the real detectable change on the measurement tool with a 95% confidence threshold [29].

Regarding validity, the analysis of convergent construct validity revealed a good association between physical functioning, role limitations caused by physical health problems, pain, and general health perceptions (PCS) in the APFD and the Arabic version of the P-QoL, as well as a good correlation between the VAS and the APFD and the P-QoL. The results of the SF-12, on the other hand, exhibited no association with those of the P-QoL, the APFD, or the VAS. This might imply that POP symptoms influence women’s physical perception rather than their emotional perception, whereas colorectal symptoms affect both physical and emotional perception. Accordingly, most of our symptomatic and asymptomatic volunteers were between the first and second stages of POP, and in these stages, symptoms may not affect their life activities and the general health perception features listed in the SF-12. When comparing the scores for the dimensions that evaluate the same symptoms, the P-QoL and APFD revealed a substantial association, which seems plausible. For example, the role limitation domain (RL) in the P-QoL covers the limitations associated with bladder and bowel problems, and as shown in Table 4, it had significant correlations (at the level of 0.01) with the Australian Pelvic Floor Dysfunction Questionnaire’s bladder function (APFD1), representing bladder symptoms, and the Australian Pelvic Floor Dysfunction Questionnaire’s bowel function (APFD2), representing bowel symptoms. Furthermore, symptomatic women had substantially higher total domain scores than asymptomatic women. The same method of evaluating the convergent construct validity was performed in the Spanish version [28], where the Pelvic Floor Impact Questionnaire (PFIQ-7) and the Pelvic Floor Distress Inventory (PFDI-20) were utilized for comparison. In contrast to the Spanish [f21] and Dutch [14] versions, in which the discriminative validity was calculated using The Mann–Whitney U test and expressed as the median (interquartile range), we used the mean percentage. This result demonstrated that there were statistically significant differences between the P-QoL, the APFD, and the VAS, at a statistically significant value of α ≤ 0.01. With higher average scores, the evidence was in favor of the symptomatic group. However, we found no statistically significant differences with the results of the SF-12, as the Mann–Whitney test results were not statistically significant.

### Limitations

This study has several limitations, including the limited number of elderly (aged 50+), who may express more symptomatic features of POP among life domains, such as role limitations, personal relationships, sleep/energy, and severity measures. Another limitation is the high heterogeneity between groups. In addition, this study did not assess the responsiveness of the P-QoL.

## 5. Conclusions

The Arabic version of the P-QoL showed a psychometric resemblance to the English version based on the significant correlations observed between the Arabic P-QoL, the Arabic APFD, and the VAS. The results revealed good reliability, internal consistency, and contrast validity. It is recommended that Arabic-speaking females with pelvic organ prolapse use the Arabic version of the P-QoL.

## 6. Recommendations

Further research is needed to assess the responsiveness of the P-QoL.

## Figures and Tables

**Figure 1 healthcare-12-00444-f001:**
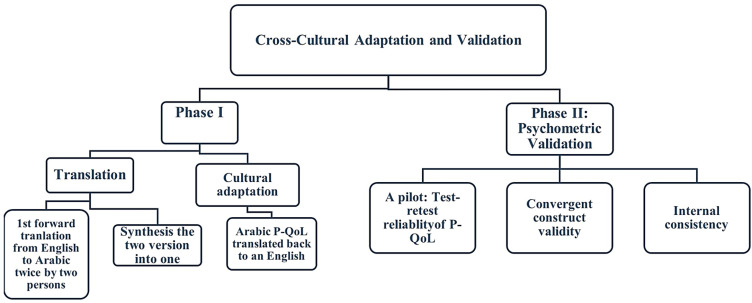
Flowchart of the two main research processes.

**Figure 2 healthcare-12-00444-f002:**
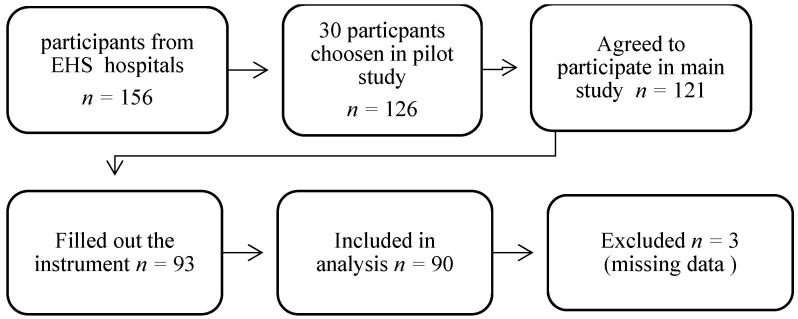
Participants’ recruitment.

**Table 1 healthcare-12-00444-t001:** Sociodemographic characteristics of the participants in the main study.

	Pilot Study	Main Study		
	*n* = 30	Symptomatic *n* = 55	Asymptomatic *n* = 35	*p*-Value
Age, *n* (%)				0.001 **
Mean (±SD)	35.9 (±11.7)	42.53 (±7.64)	33.54 (±7.12)	
18–24	3 (10.0)	0	4 (4.4)	
25–34	11 (36.6)	15 (16.7)	15 (16.7)	
35–44	10 (33.3)	14 (15.6)	14 (15.6)	
45–54	3 (10.0)	23 (25.6)	2 (2.2)	
55–64	3 (10.0)	3 (3.3)	0	
Parity, *n* (%)				
None	0 (0.0)	4 (4.4)	16 (17.8)	0.001 **
1–3	12 (40.0)	17 (18.9)	13 (14.4)	
4	4 (13.3)	30 (33.3)	2 (2.2)	
More than 4	14 (46.6)	4 (4.4)	4 (4.4)	
Number of Deliveries, Mean (±SD)				
Vaginal delivery		4.18 (±2.76)	1.40 (±1.77)	
Cesarean delivery		0.91 (±1.46)	0.60 (±1.01)	
	POP—Q stage, *n* (%)			0.001 **
Mean (±SD)		1.82 (±0.77)	0.54 (±0.51)	
Stage 0		0	16 (17.7)	
Stage 1		21 (23.3)	19 (21.1)	
Stage 2		24 (26.6)	0	
Stage 3		9 (10.0)	0	
Stage 4		1 (1.1)	0	
	Education Status Level, *n* (%)			
Preparatory school	3 (10.0)	1 (1.1)	0	0.001 **
High school graduate	15 (50.0)	52 (57.8)	5 (5.6)	
Undergraduate (university level)	12 (40.0)	2 (2.2)	30 (33.3)	
	BMI (kg/m^2^), *n* (%)			
Normal (18.5–25)	18 (60.0)	16 (17.7)	11 (12.2)	0.971
Overweight (25–30)	10 (33.3)	18 (20.0)	10 (11.1)	
Obese class I (30–35)	2 (6.6)	19 (21.1)	13 (14.4)	
Obese class II (35–40)	2 (2.2)	2 (2.2)	1 (1.1)	

SD = standard deviation, POP—Q = pelvic organ prolapse quantification system; chi-square test was used to calculate *p*-value; ** *p* < 0.001.

**Table 2 healthcare-12-00444-t002:** Test–retest reliability and internal consistency.

	Test–Retest Reliability (N = 30)		Internal Consistency (N = 90)
	ICC	*p*-Value for ICC	Cronbach’s Alpha
P-QoL	0.987	<0.001	0.971
General Health Perceptions	0.974	<0.001	
Prolapse Impact	0.969	<0.001	
Role Limitations	0.989	<0.001	0.936
Physical Limitations	0.980	<0.001	0.836
Social Limitations	0.947	<0.001	0.860
Personal Relationships	0.985	<0.001	0.896
Emotions	0.967	<0.001	0.918
Sleep/Energy	0.935	<0.001	0.682
Severity Measures	0.711	<0.001	0.885

ICC = intraclass correlation coefficient.

**Table 3 healthcare-12-00444-t003:** Reliability of the measures using standard error of measurement and minimal detectable change.

	ICC	SDdiff	SEM	MDC
P-QoL	0.987 (0.972–0.994)	5.33	0.61	1.19
General Health Perceptions	0.954 (0.903–0.978)	0.26	0.06	0.11
Prolapse Impact	0.954 (0.904–0.978)	0.37	0.08	0.16
Role Limitations	0.982 (0.962–0.991)	2.89	0.39	0.76
Physical Limitations	0.989 (0.976–0.995)	0.37	0.04	0.08
Social Limitations	0.980 (0.958–0.992)	0.78	0.11	0.22
Personal Relationships	0.934 (0.861–0.969)	1.00	0.26	0.50
Emotions	0.985 (0.968–0.993)	0.63	0.08	0.15
Sleep/Energy	0.967 (0.930–0.984)	0.57	0.10	0.20
Severity Measures	0.973 (0.944–0.987)	0.68	0.11	0.22

ICC = interclass correlation coefficient, SDdiff = standard error differences, SEM = standard error of measurement, MDC = minimum detectable change.

**Table 4 healthcare-12-00444-t004:** The analysis of convergent construct validity (Spearman’s coefficient (*r*)) for the P-QoL, APFD, SF-12, and VAS.

			P-QoL	APFD	SF-12	VAS
Spearman’s rho	P-QoL	Correlation Coefficient	1.000	0.676 **	0.020	0.471 **
		Sig. (2-tailed)	-	0.000	0.855	0.000
		N	90	90	90	89
	APFD	Correlation Coefficient	0.676 **	1.000	0.002	0.463 **
		Sig. (2-tailed)	0.000	-	0.988	0.000
		N	90	90	90	89
	SF-12	Correlation Coefficient	0.020	0.002	1.000	−0.148
		Sig. (2-tailed)	0.855	0.988	-	0.166
		N	90	90	90	89
	VAS	Correlation Coefficient	0.471 **	0.463 **	−0.148	1.000
		Sig. (2-tailed)	0.000	0.000	0.166	-
		N	89	89	89	89

** Correlation is significant at the 0.01 level (2-tailed).

**Table 5 healthcare-12-00444-t005:** The analysis of convergent construct validity (Spearman’s coefficient (*r*)) for subscales of the P-QoL and APFD, SF-12, and VAS.

	P-QoL (GH)	P-QoL (PI)	P-QoL (RL)	P-QoL (PL)	P-QoL (SL)	P-QoL (PR)	P-QoL (Emotions)	P-QoL (Sleep/Energy)	P-QoL (SM)	P-QoL
P-QoL	0.638 **	0.830 **	0.978 **	0.865 **	0.841 **	0.688 **	0.843 **	0.776 **	0.797 **	-
APFD1	0.595 **	0.695 **	0.853 **	0.790 **	0.793 **	0.637 **	0.723 **	0.680 **	0.713 **	0.873 **
APFD2	0.414 **	0.499 **	0.754 **	0.583 **	0.585 **	0.427 **	0.515 **	0.543 **	0.569 **	0.728 **
APFD3	0.211 *	0.232 *	0.288 **	0.288 **	0.312 **	0.047	0.342 **	0.218 *	0.327 **	0.297 **
APFD	0.443 **	0.492 **	0.674 **	0.595 **	0.613 **	0.371 **	0.585 **	0.494 **	0.594 **	0.677 **
SF-12	0.017	−0.091	0.049	0.075	−0.050	−0.049	0.041	−0.040	0.032	0.024
VAS	0.333 **	0.363 **	0.493 **	0.432 **	0.447 **	0.445 **	0.439 **	0.378 **	0.308 **	0.471 **

** Correlation is significant at the 0.001 level (2-tailed). * Correlation is significant at the 0.05 level (2-tailed). P-QoL = Prolapse Quality of Life Questionnaire, APFD = Australian Pelvic Floor Dysfunction Questionnaire, SF-12 = Short-Form Health Survey, VAS = Visual Analogue Scale, GH = general health perceptions, PI = prolapse impact, RL = role limitations, PL = physical limitations, SL = social limitations, PR = personal relationships, SM = severity measures, APFD1 = Australian Pelvic Floor Dysfunction Questionnaire (bladder function), APFD2 = Australian Pelvic Floor Dysfunction Questionnaire (bowel function), APFD3 = Australian Pelvic Floor Dysfunction Questionnaire (prolapse symptoms).

**Table 6 healthcare-12-00444-t006:** Discriminative validity using Mann–Whitney test.

Mean of Rank	Symptomatic (*n* = 55) (Mean%)	Asymptomatic (*n* = 35) (Mean%)	*p*-Value
P-QoL	60.46	21.99	0.001 *
APFD	54.93	30.69	0.001 *
SF-12	44.64	46.86	0.693
VAS	52.78	33	0.001 *

* *p* < 0.05.

## Data Availability

The data presented in this study are available upon request from the corresponding author. The data are not publicly available due to their use in further analysis.

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
