# Peer review of "Cross-Cultural Adaptation and Validation of the Arabic Version of the Prolapse Quality of Life Questionnaire in the United Arab Emirates"

_healthcare, 2024, doi:10.3390/healthcare12040444_

Round 1

Reviewer 1 Report

Comments and Suggestions for Authors

The manuscript by Alzarooni and colleagues validated the Arabic version of the P-QoL questionnaire. Overall the manuscript is well-written and this is an interesting study. I have a few comments that I think could help strengthen the presentation of the methods and results.

  • Please spell out the abbreviations before using it for the first time. For example, SF-12 in line 39.
  • In the participants section, should speaking or reading Arabic be one of the inclusion criteria?
  • In 2.6 statistical analysis, please indicate two-sided p-values < 0.05 were used to determine statistical significance. Please also describe what statistical analyses were used to compare the sociodemographic characteristics of the participants in Table 1.
  • In line 167, please elaborate on the descriptive statistics that were used for sociodemographic features and specific clinical background data. For example, mean and standard deviation were used to describe continuous variables. Frequency and proportion were used to describe categorical variables.
  • In line 150, you mentioned that a pilot was conducted on randomly chosen 30 participants who have completed the Arabic P-QoL 2 times with a time gap of at least 48 h to evaluate test-retest reliability. Please elaborate on how you recruited these 30 participants and present their demographics in a table. Please also compare the demographics of these 30 participants vs. the 90 participants included in the other analysis and show if there is any significant difference.

Author Response

Dear Reviewer,

We are grateful to the reviewers for their insightful comments on our paper. We have been able to incorporate changes to reflect most of the suggestions provided by the reviewers. We have highlighted the changes within the manuscript.

Here is a point-by-point response to the reviewers’ comments and concerns.

Comment 1: Please spell out the abbreviations before using it for the first time. For example, SF-12 in line 39.

Response: The full term of 12 Items Short form survey has been added in the mention line.

Comment 2: In the participants section, should speaking or reading Arabic be one of the inclusion criteria?

Response: Yes, it’s an inclusion criterion, participants should speak and read Arabic. Please see line 108.

Comment 3: In 2.6 statistical analysis, please indicate two-sided p-values < 0.05 were used to determine statistical significance. Please also describe what statistical analyses were used to compare the sociodemographic characteristics of the participants in Table 1.

Response: Two-sided p-values < 0.05 was add to 2.6 section, Baseline demographic data were analyzed by Fischer’ exact test

Comment 4: In line 167, please elaborate on the descriptive statistics that were used for sociodemographic features and specific clinical background data. For example, mean and standard deviation were used to describe continuous variables. Frequency and proportion were used to describe categorical variables.

Response: In 2.6 statistical analysis which is line 170, Descriptive statistics were performed to find out mean and standard deviation for sociodemographic features and specific clinical background data for continues variables including Age, Pelvic Organ Prolapse Quantification system (POP—Q stages) and Type of Delivery, while frequency and proportion were used to describe categorical variables including Parity, Education status level and Body mass Index (BMI).

Comment 5: In line 150, you mentioned that a pilot was conducted on randomly chosen 30 participants who have completed the Arabic P-QoL 2 times with a time gap of at least 48 h to evaluate test-retest reliability. Please elaborate on how you recruited these 30 participants and present their demographics in a table. Please also compare the demographics of these 30 participants vs. the 90 participants included in the other analysis and show if there is any significant difference.

Response: According to the reviewer’s recommendation, recruitment of the pilot study has been elaborated further from line 150. The 30 participants with the help of simple random sampling by lottery method. Pilot study participants’ information has been added in Table 1 separately from main study participants. They were not included in the main study which they were not categorized as two groups.

Additional clarifications

In addition to the above comments, figure 2 has been re-designed.

Reviewer 2 Report

Comments and Suggestions for Authors

The manuscript submitted for review concerns a very important element: women's quality of life.

The manuscript concerns an Arabic linguistic and cultural adaptation of a questionnaire assessing the quality of life of women with female genital prolapse.

The quality of life of women, also in Arab countries, is very important, interesting, and worthy of scientific research. I believe that the research was conducted by a team that also included women.

The manuscript is ready for publication after a few corrections.

1. Authors sometimes refer to the quality of life as QoL and sometimes as QOL. This requires standardization. I suggest QoL. (l. 49)

2. The authors state that the urinary bladder and rectum are reproductive organs. It's not true. (l.56,57)

3. In phase I there were two steps, not two phases (l.143,147)

4. Table 1. Contains illegible column headings, e.g.: Mean above Age and Age above Mean, in what unit authors report “Delivery”, what is in brackets. This table needs to be thoroughly revised.

5. What is this abbreviation (SDdiff)? Shouldn't it be - SEdiff

6. Emotions, Sleep/Energy abbreviation (SE) stands for Standard Error. I propose to change it (l.231 and Table 5)

7. Why is the Mann-Whitney test in quotation marks (l.247,248)

7. In the Discussion, the authors write "previous studies" without providing a literature source (l.275)

8. The discussion is largely a repetition of the results and does not compare them with the literature. Needs improvement.

Author Response

Comment 1: Authors sometimes refer to the quality of life as QoL and sometimes as QOL. This requires standardization. I suggest QoL. (l. 49)

Response:  Thank you for your comment. QOL replaced by QoL as your suggestion.

Comment 2: The authors state that the urinary bladder and rectum are reproductive organs. It's not true. (l.56,57)

Response: line 56.57 reproductive organs replaced by pelvic organs.

Comment 3: In phase I there were two steps, not two phases (l.143,147).

Response: Totally agree with you, line 144, 148 have been correct from two phases to two steps.

Comment 4: Table 1. Contains illegible column headings, e.g.: Mean above Age and Age above Mean, in what unit authors report “Delivery”, what is in brackets. This table needs to be thoroughly revised.

Response: table 1 has been revised and correction done and highlighted. Delivery has been changed to number of deliveries.

Comment 5: What is this abbreviation (SDdiff)? Shouldn't it be – Sediff

Response: SDdiff is refers to the standard deviation of the differences.

Comment 6: Emotions, Sleep/Energy abbreviation (SE) stands for Standard Error. I propose to change it (l.231 and Table 5)

Response: In line 231 and Table 5 abbreviation have been removed to avoid confusion.

Comment 7: Why is the Mann-Whitney test in quotation marks (l.247,248)

Response: Quotation marks has been removed.

Comment 8: In the Discussion, the authors write "previous studies" without providing a literature source (l.275)

Response: The previous studies were mentioned before in line 63, 64 ,65.

(To accomplish this, in 2003, condition specific QoL instruments were created and published in Italy by Digesu et al. [6]. It was then translated into several languages, including English, German, Dutch, Slovakian, Persian, Portuguese, Thai, Japanese, Amharic, and Turkish [4,6–14].)

Comment 9: The discussion is largely a repetition of the results and does not compare them with the literature. Needs improvement.

Response: Thank you for your comment. The maximum literature support related to the study have been included in the discussion part.

Additional clarifications

In addition to the above comments, figure 2 has been re-designed.

Reviewer 3 Report

Comments and Suggestions for Authors

Dear Author,

It is a well-known scale. It is a well-written manuscript. Bu t I have some comments.

You should add an explanation about sample size calculation.

You should add an explanation about sample selection process.

You should add an explanation confirmatory and explanatory factor analysis.

You should check your analysis about chi-square. It may be Fischer exact test, or it is not suitable for chi-square analysis.

You should add an explanations about sample size for each stage. How many people participate in each stage?  How many people participate pilot study?

You should do confirmatory factor analysis. 

You should give KMO, X2/df, p value, CFI, GFI, and RMSEA values.

You may follow ITC guideline or COSMIN guideline or WHO guideline for reporting.

You should follow international guideline reporting process.

Best wishes

Comments on the Quality of English Language

Minor editing of English language required.

Author Response

Dear Reviewer,

We are grateful to the reviewers for their insightful comments on our paper. We have been able to incorporate changes to reflect most of the suggestions provided by the reviewers. We have highlighted the changes within the manuscript.

Here is a point-by-point response to the reviewers’ comments and concerns.

Comment 1: You should add an explanation about sample size calculation.

Response: Our main outcome was evaluated by Cronbach’s alpha. 10% was added considering a possible dropout. An explanation has been added in the corresponding section.

Comment 2: You should add an explanation about sample selection process.

Response: In line 150, 151 sample selection process added for more clarifying as requested and highlighted.

Comment 3: You should add an explanation confirmatory and explanatory factor analysis.

Response: We really appreciate this valid point. In line 258, CFI value have been added showing accepted fit.

Comment 4: You should check your analysis about chi-square. It may be Fischer exact test, or it is not suitable for chi-square analysis.

Response: Fischer exact test done to calculate p value which result in significant (p value) 0.001.

Comment 5: You should add an explanation about sample size for each stage. How many people participate in each stage?  How many people participate pilot study?

Response: In line 150, recruitment had been elaborate throughout the full process. The 30 participants with the help of simple random sampling by lottery method. Pilot study participants were not included in the main study which they were not categorized as two groups. Please find Figure 2 presents the details of sample size for pilot and main study.

Comment 6: You should do confirmatory factor analysis

Response: We believe CFI calculation (value of 0.74) works for the analysis and included in our study.

Comment 7: You should give KMO, X2/df, p value, CFI, GFI, and RMSEA values.

Response: In line 258, 259 ,260, 261 all statistical analysis requested has been added. We believe confirmatory factor analysis CFI is 0.74, which indicates accepted fit. Kaiser-Meyer-Olkin Measure of Sampling Adequacy KMO value is 0.869, which indicate the sampling is adequate with p value 0.000. X2/df is 2.29 and the root mean square error of approximation RMSEA 0.12 which is mediocre.

Comment 8: You may follow ITC guideline or COSMIN guideline or WHO guideline for reporting.

Response: 

Thank you for your suggestion and comments. This manuscript followed the Reporting guideline that is internationally accepted as follows.

Beaton DE, Bombardier C, Guillemin F, Ferraz MB. Guidelines for the process of cross-cultural adaptation of self-report measures. Spine (Phila Pa 1976). 2000;25(24):3186–91. pmid:11124735

Comment 9: You should follow international guideline reporting process.

Response: This manuscript follows the above protocol as mentioned in comment 8.

Additional clarifications

In addition to the above comments, figure 2 has been re-designed.

Regarding the explanatory factor analysis, we were unable to figure it out within the time frame limits.

Sincerely,

Round 2

Reviewer 3 Report

Comments and Suggestions for Authors

The manuscript fulfils all the scientific standards necessary to be published.  The manuscript has the sufficient quality and originality to be published in Healthcare. I do not think that the manuscript needs further revisions. I think it can be considered for publication in journal.

Respectfully yours,